# On Communication Cost of Distributed Statistical Estimation and Dimensionality

**Ankit Garg**
Department of Computer Science, Princeton University
garg@cs.princeton.edu

**Tengyu Ma**
Department of Computer Science, Princeton University
tengyu@cs.princeton.edu

**Huy L. Nguyễn**
Simons Institute, UC Berkeley
hlnguyen@cs.princeton.edu

## Abstract

We explore the connection between dimensionality and communication cost in distributed learning problems. Specifically we study the problem of estimating the mean $\vec{\theta}$ of an unknown $d$ dimensional gaussian distribution in the distributed setting. In this problem, the samples from the unknown distribution are distributed among $m$ different machines. The goal is to estimate the mean $\vec{\theta}$ at the optimal minimax rate while communicating as few bits as possible. We show that in this setting, the communication cost scales linearly in the number of dimensions i.e. one needs to deal with different dimensions individually. Applying this result to previous lower bounds for one dimension in the interactive setting [1] and to our improved bounds for the simultaneous setting, we prove new lower bounds of $\Omega(md/\log(m))$ and $\Omega(md)$ for the bits of communication needed to achieve the minimax squared loss, in the interactive and simultaneous settings respectively. To complement, we also demonstrate an interactive protocol achieving the minimax squared loss with $O(md)$ bits of communication, which improves upon the simple simultaneous protocol by a logarithmic factor. Given the strong lower bounds in the general setting, we initiate the study of the distributed parameter estimation problems with structured parameters. Specifically, when the parameter is promised to be $s$-sparse, we show a simple thresholding based protocol that achieves the same squared loss while saving a $d/s$ factor of communication. We conjecture that the tradeoff between communication and squared loss demonstrated by this protocol is essentially optimal up to logarithmic factor.

## 1  Introduction

The last decade has witnessed a tremendous growth in the amount of data involved in machine learning tasks. In many cases, data volume has outgrown the capacity of memory of a single machine and it is increasingly common that learning tasks are performed in a distributed fashion on many machines. Communication has emerged as an important resource and sometimes the bottleneck of the whole system. A lot of recent works are devoted to understand how to solve problems distributedly with efficient communication [2, 3, 4, 1, 5].

In this paper, we study the relation between the *dimensionality* and the communication cost of statistical estimation problems. Most modern statistical problems are characterized by high dimensionality. Thus, it is natural to ask the following meta question:

*How does the communication cost scale in the dimensionality?*

We study this question via the problems of estimating parameters of distributions in the distributed setting. For these problems, we answer the question above by providing two complementary results:

1. Lower bound for general case: If the distribution is a product distribution over the coordinates, then one essentially needs to estimate each dimension of the parameter individually and the information cost (a proxy for communication cost) scales linearly in the number of dimensions.

2. Upper bound for sparse case: If the true parameter is promised to have low sparsity, then a very simple thresholding estimator gives better tradeoff between communication cost and mean-square loss.

Before getting into the ideas behind these results, we first define the problem more formally. We consider the case when there are $m$ machines, each of which receives $n$ i.i.d samples from an unknown distribution $P$ (from a family $\mathcal{P}$) over the $d$-dimensional Euclidean space $\mathbb{R}^d$. These machines need to estimate a parameter $\theta$ of the distribution via communicating with each other. Each machine can do arbitrary computation on its samples and messages it receives from other machines. We regard communication (the number of bits communicated) as a resource, and therefore we not only want to optimize over the estimation error of the parameters but also the tradeoff between the estimation error and communication cost of the whole procedure. For simplicity, here we are typically interested in achieving the minimax error [1] while communicating as few bits as possible. Our main focus is the high dimensional setting where $d$ is very large.

**Communication Lower Bound via Direct-Sum Theorem**   The key idea for the lower bound is, when the unknown distribution $P = P_1 \times \cdots \times P_d$ is a product distribution over $\mathbb{R}^d$, and each coordinate of the parameter $\theta$ only depends on the corresponding component of $P$, then we can view the $d$-dimensional problem as $d$ independent copies of one dimensional problem. We show that, one unfortunately cannot do anything beyond this trivial decomposition, that is, treating each dimension independently, and solving $d$ different estimations problems individually. In other words, the communication cost [2] must be at least $d$ times the cost for one dimensional problem. We call this theorem "direct-sum" theorem.

To demonstrate our theorem, we focus on the specific case where $P$ is a $d$ dimensional spherical Gaussian distribution with an unknown mean and covariance $\sigma^2 I_d$ [3] . The problem is to estimate the mean of $P$. The work [1] showed a lower bound on the communication cost for this problem when $d = 1$. Our technique when applied to their theorem immediately yields a lower bound equal to $d$ times the lower bound for the one dimension problem for any choice of $d$. Note that [5] independently achieve the same bound by refining the proof in [1].

In the simultaneous communication setting, where all machines send one message to one machine and this machine needs to figure out the estimation, the work [1] showed that $\Omega(md/\log m)$ bits of communication are needed to achieve the minimax squared loss. In this paper, we improve this bound to $\Omega(md)$, by providing an improved lower bound for one-dimensional setting and then applying our direct-sum theorem.

The direct-sum theorem that we prove heavily uses the idea and tools from the recent developments in communication complexity and information complexity. There has been a lot of work on the paradigm of studying communication complexity via the notion of information complexity [6, 7, 8, 9, 10]. Information complexity can be thought of as a proxy for communication complexity that is especially accurate for solving multiple copies of the same problem simultaneously [8]. Proving so-called "direct-sum" results has become a standard tool, namely the fact that the amount of resources required for solving $d$ copies of a problem (with different inputs) in parallel is equal to $d$ times the amount required for one copy. In other words, there is no saving from solving many copies of the same problem in batch and the trivial solution of solving each of them separately is optimal. Note that this generic statement is certainly NOT true for arbitrary type of tasks and arbitrary type of resources. Actually even for distributed computing tasks, if the measure of resources is the

communication cost instead of information cost, there exist examples where solving $d$ copies of a certain problem requires less communication than $d$ times the communication required for one copy [11]. Therefore, a direct-sum theorem, if true, could indeed capture the features and difficulties of the problems.

Our result can be viewed as a direct sum theorem for communication complexity for statistical estimation problems: the amount of communication needed for solving an estimation problem in $d$ dimensions is at least $d$ times the amount of information needed for the same problem in one dimension. The proof technique is directly inspired by the notion of conditional information complexity [7], which was used to prove direct sum theorems and lower bounds for streaming algorithms. We believe this is a fruitful connection and can lead to more lower bounds in statistical machine learning.

To complement the above lower bounds, we also show an interactive protocol that uses a log factor less communication than the simple protocol, under which each machine sends the sample mean and the center takes the average as the estimation. Our protocol demonstrates additional power of interactive communication and potential complexity of proving lower bound for interactive protocols.

**Thresholding Algorithm for Sparse Parameter Estimation**    In light of the strong lower bounds in the general case, a question suggests itself as a way to get around the impossibility results:

*Can we do better when the data (parameters) have more structure?*

We study this questions by considering the sparsity structure on the parameter $\theta$. Specifically, we consider the case when the underlying parameter $\theta$ is promised to be $s$-sparse. We provide a simple protocol that achieves the same squared-loss $O(d\sigma^2/(mn))$ as in the general case, while using $\tilde{O}(sm)$ communications, or achieving optimal squared loss $O(s\sigma^2/(mn))$, with communication $\tilde{O}(dm)$, or any tradeoff between these cases. We even conjecture that this is the best tradeoff up to polylogarithmic factors.

## 2   Problem Setup, Notations and Preliminaries

**Classical Statistical Parameter Estimation** We start by reviewing the classical framework of statistical parameter estimation problems. Let $\mathcal{P}$ be a family of distributions over $\mathcal{X}$. Let $\theta : \mathcal{P} \rightarrow \Theta \subset \mathbb{R}$ denote a function defined on $\mathcal{P}$. We are given samples $X^1, \ldots, X^n$ from some $P \in \mathcal{P}$, and are asked to estimate $\theta(P)$. Let $\hat{\theta} : \mathcal{X}^n \rightarrow \Theta$ be such an estimator, and $\hat{\theta}(X^1, \ldots, X^n)$ is the corresponding estimate.

Define the squared loss $R$ of the estimator to be

$$R(\hat{\theta}, \theta) = \mathop{\mathbb{E}}_{\hat{\theta}, X} \left[ \|\hat{\theta}(X^1, \ldots, X^n) - \theta(P)\|_2^2 \right]$$

In the high-dimensional case, let $\mathcal{P}^d := \{\vec{P} = P_1 \times \cdots \times P_d : P_i \in \mathcal{P}\}$ be the family of product distributions over $\mathcal{X}^d$. Let $\vec{\theta} : \mathcal{P}^d \rightarrow \Theta^d \subset \mathbb{R}^d$ be the $d$-dimensional function obtained by applying $\theta$ point-wise $\vec{\theta}(P_1 \times \cdots \times P_d) = (\theta(P_1), \ldots, \theta(P_d))$.

Throughout this paper, we consider the case when $\mathcal{X} = \mathbb{R}$ and $\mathcal{P} = \{\mathcal{N}(\theta, \sigma^2) : \theta \in [-1, 1]\}$ is Gaussian distribution with for some fixed and known $\sigma$. Therefore, in the high-dimensional case, $\mathcal{P}^d = \{\mathcal{N}(\vec{\theta}, \sigma^2 I_d) : \vec{\theta} \in [-1, 1]^d\}$ is a collection of spherical Gaussian distributions. We use $\hat{\vec{\theta}}$ to denote the $d$-dimensional estimator. For clarity, in this paper, we always use $\vec{\cdot}$ to indicate a vector in high dimensions.

**Distributed Protocols and Parameter Estimation:**  In this paper, we are interested in the situation where there are $m$ machines and the $j$th machine receives $n$ samples $\vec{X}^{(j,1)}, \ldots, \vec{X}^{(j,n)} \in \mathbb{R}^d$ from the distribution $\vec{P} = \mathcal{N}(\vec{\theta}, \sigma^2 I_d)$. The machines communicate via a publicly shown blackboard. That is, when a machine writes a message on the blackboard, all other machines can see the content of the message. Following [1], we usually refer to the blackboard as the *fusion center* or simply *center*. Note that this model captures both point-to-point communication as well as broadcast com-

munication. Therefore, our lower bounds in this model apply to both the message passing setting and the broadcast setting. We will say that a protocol is simultaneous if each machine broadcasts a single message based on its input independently of the other machine ([1] call such protocols independent).

We denote the collection of all the messages written on the blackboard by $Y$. We will refer to $Y$ as transcript and note that $Y \in \{0,1\}^*$ is written in bits and the communication cost is defined as the length of $Y$, denoted by $|Y|$. In multi-machine setting, the estimator $\hat{\vec{\theta}}$ only sees the transcript $Y$, and it maps $Y$ to $\hat{\vec{\theta}}(Y)$ [4], which is the estimation of $\vec{\theta}$. Let letter $j$ be reserved for index of the machine and $k$ for the sample and letter $i$ for the dimension. In other words, $\vec{X}_i^{(j,k)}$ is the $i$th-coordinate of $k$th sample of machine $j$. We will use $\vec{X}_i$ as a shorthand for the collection of the $i$th coordinate of all the samples: $\vec{X}_i = \{\vec{X}_i^{(j,k)} : j \in [m], k \in [n]\}$. Also note that $[n]$ is a shorthand for $\{1, \ldots, n\}$.

The mean-squared loss of the protocol $\Pi$ with estimator $\hat{\vec{\theta}}$ is defined as

$$R\left((\Pi, \hat{\vec{\theta}}), \vec{\theta}\right) = \sup_{\vec{\theta}} \, \mathbb{E}_{\vec{X}, \Pi} [\|\hat{\vec{\theta}}(Y) - \vec{\theta}\|^2]$$

and the communication cost of $\Pi$ is defined as

$$\mathrm{CC}(\Pi) = \sup_{\vec{\theta}} \, \mathbb{E}_{\vec{X}, \Pi} [|Y|]$$

The main goal of this paper is to study the tradeoff between $R\left((\Pi, \hat{\vec{\theta}}), \vec{\theta}\right)$ and $\mathrm{CC}(\Pi)$.

**Proving Minimax Lower Bound:** We follow the standard way to prove minimax lower bound. We introduce a (product) distribution $\mathcal{V}^d$ of $\vec{\theta}$ over the $[-1,1]^d$. Let's define the mean-squared loss with respect to distribution $\mathcal{V}^d$ as

$$R_{\mathcal{V}^d}((\Pi, \hat{\vec{\theta}}), \vec{\theta}) = \mathbb{E}_{\vec{\theta} \sim \mathcal{V}^d} \left[ \mathbb{E}_{\vec{X}, \Pi} [\|\hat{\vec{\theta}}(Y) - \vec{\theta}\|^2] \right]$$

It is easy to see that $R_{\mathcal{V}^d}((\Pi, \hat{\vec{\theta}}), \vec{\theta}) \leq R((\Pi, \hat{\vec{\theta}}), \vec{\theta})$ for any distribution $\mathcal{V}^d$. Therefore to prove lower bound for the minimax rate, it suffices to prove the lower bound for the mean-squared loss under any distribution $\mathcal{V}^d$. [5]

**Private/Public Randomness:** We allow the protocol to use both private and public randomness. Private randomness, denoted by $R_{\mathrm{priv}}$, refers to the random bits that each machine draws by itself. Public randomness, denoted by $R_{\mathrm{pub}}$, is a sequence of random bits that is shared among all parties before the protocol without being counted toward the total communication. Certainly allowing these two types of randomness only makes our lower bound stronger, and public randomness is actually only introduced for convenience.

Furthermore, we will see in the proof of Theorem 3.1, the benefit of allowing private randomness is that we can hide information using private randomness when doing the reduction from one dimension protocol to $d$-dimensional one. The downside is that we require a stronger theorem (that tolerates private randomness) for the one dimensional lower bound, which is not a problem in our case since technique in [1] is general enough to handle private randomness.

**Information cost:** We define information cost $\mathrm{IC}(\Pi)$ of protocol $\Pi$ as mutual information between the data and the messages communicated conditioned on the mean $\vec{\theta}$. [6]

$$\text{IC}_{\mathcal{V}^d}(\Pi) = I(\vec{X}; Y \mid \vec{\theta}, R_{\text{pub}})$$

Private randomness doesn't explicitly appear in the definition of information cost but it affects it. Note that the information cost is a lower bound on the communication cost:

$$\text{IC}_{\mathcal{V}^d}(\Pi) = I(\vec{X}; Y \mid \vec{\theta}, R_{\text{pub}}) \leq H(Y) \leq \text{CC}(\Pi)$$

The first inequality uses the fact that $I(U; V \mid W) \leq H(V \mid W) \leq H(V)$ hold for any random variable $U, V, W$, and the second inequality uses Shannon's source coding theorem [13].

We will drop the subscript for the prior $\mathcal{V}^d$ of $\vec{\theta}$ when it is clear from the context.

## 3 Main Results

### 3.1 High Dimensional Lower bound via Direct Sum

Our main theorem roughly states that if one can solves the $d$-dimensional problem, then one must be able to solve the one dimensional problem with information cost and square loss reduced by a factor of $d$. Therefore, a lower bound for one dimensional problem will imply a lower bound for high dimensional problem, with information cost and square loss scaled up by a factor of $d$.

We first define our task formally, and then state the theorem that relates $d$-dimensional task with one-dimensional task.

**Definition 1.** We say a protocol and estimator pair $(\Pi, \hat{\vec{\theta}})$ solves task $T(d, m, n, \sigma^2, \mathcal{V}^d)$ with information cost $C$ and mean-squared loss $R$, if for $\vec{\theta}$ randomly chosen from $\mathcal{V}^d$, $m$ machines, each of which takes $n$ samples from $\mathcal{N}(\vec{\theta}, \sigma^2 I_d)$ as input, can run the protocol $\Pi$ and get transcript $Y$ so that the followings are true:

$$R_{\mathcal{V}^d}((\Pi, \hat{\vec{\theta}}), \vec{\theta}) = R \tag{1}$$
$$I_{\mathcal{V}^d}(\vec{X}; Y \mid \vec{\theta}, R_{\text{pub}}) = C \tag{2}$$

**Theorem 3.1.** *[Direct-Sum] If $(\Pi, \hat{\vec{\theta}})$ solves the task $T(d, m, n, \sigma^2, \mathcal{V}^d)$ with information cost $C$ and squared loss $R$, then there exists $(\Pi', \hat{\theta})$ that solves the task $T(1, m, n, \sigma^2, \mathcal{V})$ with information cost at most $4C/d$ and squared loss at most $4R/d$. Furthermore, if the protocol $\Pi$ is simultaneous, then the protocol $\Pi'$ is also simultaneous.*

**Remark 1.** Note that this theorem doesn't prove directly that communication cost scales linearly with the dimension, but only information cost. However for many natural problems, communication cost and information cost are similar for one dimension (e.g. for gaussian mean estimation) and then this direct sum theorem can be applied. In this sense it is very generic tool and is widely used in communication complexity and streaming algorithms literature.

**Corollary 3.1.** *Suppose $(\Pi, \hat{\vec{\theta}})$ estimates the mean of $\mathcal{N}(\vec{\theta}, \sigma^2 I_d)$, for all $\vec{\theta} \in [-1, 1]^d$, with mean-squared loss $R$, and communication cost $B$. Then*

$$R \geq \Omega\left(\min\left\{\frac{d^2\sigma^2}{nB\log m}, \frac{d\sigma^2}{n\log m}, d\right\}\right)$$

*As a corollary, when $\sigma^2 \leq mn$, to achieve the mean-squared loss $R = \frac{d\sigma^2}{mn}$, the communication cost $B$ is at least $\Omega\left(\frac{dm}{\log m}\right)$.*

This lower bound is tight up to polylogarithmic factors. In most of the cases, roughly $B/m$ machines sending their sample mean to the fusion center and $\hat{\vec{\theta}}$ simply outputs the mean of the sample means with $O(\log m)$ bits of precision will match the lower bound up to a multiplicative $\log^2 m$ factor. [7]

## 3.2 Protocol for sparse estimation problem

In this section we consider the class of gaussian distributions with sparse mean: $\mathcal{P}_s = \{\mathcal{N}(\vec{\theta}, \sigma^2 I_d) : |\vec{\theta}|_0 \leq s, \vec{\theta} \in \mathbb{R}^d\}$. We provide a protocol that exploits the sparse structure of $\vec{\theta}$.

---

**Inputs** : Machine $j$ gets samples $X^{(j,1)}, \ldots, X^{(j,n)}$ distributed according to $\mathcal{N}(\vec{\theta}, \sigma^2 I_d)$, where $\vec{\theta} \in \mathbb{R}^d$ with $|\vec{\theta}|_0 \leq s$.

For each $1 \leq j \leq m' = (Lm \log d)/\alpha$, (where $L$ is a sufficiently large constant), machine $j$ sends its sample mean $\bar{X}^{(j)} = \frac{1}{n}\left(X^{(j,1)}, \ldots, X^{(j,n)}\right)$ (with precision $O(\log m)$) to the center.

Fusion center calculates the mean of the sample means $\bar{X} = \frac{1}{m'}\left(\bar{X}^{(1)} + \cdots + \bar{X}^{(m')}\right)$.

Let $\hat{\theta}_i = \begin{cases} \bar{X}_i & \text{if } |\bar{X}_i|^2 \geq \frac{\alpha\sigma^2}{mn} \\ 0 & \text{otherwise} \end{cases}$

**Outputs** $\hat{\vec{\theta}}$

---
**Protocol 1:** Protocol for $\mathcal{P}_s$

**Theorem 3.2.** *For any $P \in \mathcal{P}_s$, for any $d/s \geq \alpha \geq 1$, Protocol 1 returns $\vec{\theta}$ with mean-squared loss $O(\frac{\alpha s \sigma^2}{mn})$ with communication cost $O((dm \log m \log d)\alpha)$.*

The proof of the theorem is deferred to supplementary material. Note that when $\alpha = 1$, we have a protocol with $\tilde{O}(dm)$ communication cost and mean-squared loss $O(s\sigma^2/(mn))$, and when $\alpha = d/s$, the communication cost is $\tilde{O}(sm)$ but squared loss $O(d\sigma^2/(mn))$. Comparing to the case where we don't have sparse structure, basically we either replace the $d$ factor in the communication cost by the intrinsic dimension $s$ or the $d$ factor in the squared loss by $s$, but not both.

## 3.3 Improved upper bound

The lower bound provided in Section 3.1 is only tight up to polylogarithmic factor. To achieve the centralized minimax rate $\frac{\sigma^2 d}{mn}$, the best existing upper bound of $O(dm \log(m))$ bits of communication is achieved by the simple protocol that ask each machine to send its sample mean with $O(\log n)$ bits precision . We improve the upper bound to $O(dm)$ using the interactive protocols.

Recall that the class of unknown distributions of our model is $\mathcal{P}^d = \{\mathcal{N}(\vec{\theta}, \sigma^2 I_d) : \theta \in [-1,1]^d\}$.

**Theorem 3.3.** *Then there is an interactive protocol $\Pi$ with communication $O(md)$ and an estimator $\hat{\vec{\theta}}$ based on $\Pi$ which estimates $\vec{\theta}$ up to a squared loss of $O(\frac{d\sigma^2}{mn})$.*

**Remark 2.** Our protocol is interactive but not simultaneous, and it is a very interesting question whether the upper bound of $O(dm)$ could be achieved by a simultaneous protocol.

## 3.4 Improved lower bound for simultaneous protocols

Although we are not able to prove $\Omega(dm)$ lower bound for achieve the centralized minimax rate in the interactive model, the lower bound for simultaneous case can be improved to $\Omega(dm)$. Again, we lowerbound the information cost for the one dimensional problem first, and applying the direct-sum theorem in Section 3.1, we got the $d$-dimensional lower bound.

**Theorem 3.4.** *Suppose simultaneous protocol $(\Pi, \hat{\vec{\theta}})$ estimates the mean of $\mathcal{N}(\vec{\theta}, \sigma^2 I_d)$, for all $\vec{\theta} \in [-1,1]^d$, with mean-squared loss R, and communication cost B, Then*

$$R \geq \Omega\left(\min\left\{\frac{d^2\sigma^2}{nB}, d\right\}\right)$$

*As a corollary, when $\sigma^2 \leq mn$, to achieve mean-squared loss $R = \frac{d\sigma^2}{mn}$, the communication cost B is at least $\Omega(dm)$.*

# 4  Proof sketches

## 4.1  Proof sketch of theorem 3.1 and corollary 3.1

To prove a lower bound for the $d$ dimensional problem using an existing lower bound for one dimensional problem, we demonstrate a reduction that uses the (hypothetical) protocol $\Pi$ for $d$ dimensions to construct a protocol for the one dimensional problem.

For each fixed coordinate $i \in [d]$, we design a protocol $\Pi_i$ for the one-dimensional problem by embedding the one-dimensional problem into the $i^{th}$ coordinate of the $d$-dimensional problem. We will show essentially that if the machines first collectively choose randomly a coordinate $i$, and run protocol $\Pi_i$ for the one-dimensional problem, then the information cost and mean-squared loss of this protocol will be only $1/d$ factor of those of the $d$-dimensional problem. Therefore, the information cost of the $d$-dimensional problem is at least $d$ times the information cost of one-dimensional problem.

---

Inputs : Machine $j$ gets samples $X^{(j,1)}, \ldots, X^{(j,n)}$ distributed according to $\mathcal{N}(\theta, \sigma^2)$, where $\theta \sim \mathcal{V}$.

  1. All machines publicly sample $\breve{\theta}_{-i}$ distributed according to $\mathcal{V}^{d-1}$.

  2. Machine $j$ privately samples $\breve{X}_{-i}^{(j,1)}, \ldots, \breve{X}_{-i}^{(j,n)}$ distributed according to $\mathcal{N}(\breve{\theta}_{-i}, \sigma^2 I_{d-1})$. Let $\breve{X}^{(j,k)} = (\breve{X}_1^{(j,k)}, \ldots, \breve{X}_{i-1}^{(j,k)}, X^{(j,k)}, \breve{X}_{i+1}^{(j,k)}, \ldots, \breve{X}_d^{(j,k)})$.

  3. All machines run protocol $\Pi$ on data $\breve{X}$ and get transcript $Y_i$. The estimator $\hat{\theta}_i$ is $\hat{\theta}_i(Y_i) = \hat{\vec{\theta}}(Y)_i$ i.e. the $i^{th}$ coordinate of the $d$-dimensional estimator.

**Protocol 2:** $\Pi_i$

---

In more detail, under protocol $\Pi_i$ (described formally in Protocol 2) the machines prepare a $d$-dimensional dataset as follows: First they fill the one-dimensional data that they got into the $i^{th}$ coordinate of the $d$-dimensional data. Then the machines choose publicly randomly $\vec{\theta}_{-i}$ from distribution $\mathcal{V}^{d-1}$, and draw independently and privately gaussian random variables from $\mathcal{N}(\vec{\theta}_{-i}, I_{d-1})$, and fill the data into the other $d-1$ coordinates. Then machines then simply run the $d$-dimension protocol $\Pi$ on this tailored dataset. Finally the estimator, denoted by $\hat{\theta}_i$, outputs the $i^{th}$ coordinate of the $d$-dimensional estimator $\hat{\vec{\theta}}$.

We are interested in the mean-squared loss and information cost of the protocol $\Pi_i$'s that we just designed. The following lemmas relate $\Pi_i$'s with the original protocol $\Pi$.

**Lemma 1.** *Protocols $\Pi_i$'s satisfy* $\sum_{i=1}^{d} R_{\mathcal{V}}\left( (\Pi_i, \hat{\theta}_i), \theta \right) = R_{\mathcal{V}^d}\left( (\Pi, \hat{\vec{\theta}}), \vec{\theta} \right)$

**Lemma 2.** *Protocols $\Pi_i$'s satisfy* $\sum_{i=1}^{d} IC_{\mathcal{V}}(\Pi_i) \leq IC_{\mathcal{V}^d}(\Pi)$

Note that the counterpart of Lemma 2 with communication cost won't be true, and actually the communication cost of each $\Pi_i$ is the same as that of $\Pi$. It turns out doing reduction in communication cost is much harder, and this is part of the reason why we use information cost as a proxy for communication cost when proving lower bound. Also note that the correctness of Lemma 2 heavily relies on the fact that $\Pi_i$ draws the redundant data privately independently (see Section 2 and the proof for more discussion on private versus public randomness).

By Lemma 1 and Lemma 2 and a Markov argument, there exists an $i \in \{1, \ldots, d\}$ such that

$$R\left( (\Pi_i, \hat{\theta}_i), \theta \right) \leq \frac{4}{d} \cdot R\left( (\Pi, \vec{\theta}), \vec{\theta} \right) \quad \text{and} \quad \text{IC}(\Pi_i) \leq \frac{4}{d} \cdot \text{IC}(\Pi)$$

Then the pair $(\Pi', \hat{\theta}) = (\Pi_i, \hat{\theta}_i)$ solves the task $T(1, m, n, \sigma^2, \mathcal{V})$ with information cost at most $4C/d$ and squared loss $4R/d$, which proves Theorem 3.1.

Corollary 3.1 follows Theorem 3.1 and the following lower bound for one dimensional gaussian mean estimation proved in [1]. We provide complete proofs in the supplementary.

**Theorem 4.1.** *[1] Let $\mathcal{V}$ be the uniform distribution over $\{\pm\delta\}$, where $\delta^2 \leq \min\left(1, \frac{\sigma^2 \log(m)}{n}\right)$. If $(\Pi, \hat{\theta})$ solves the task $T(1, m, n, \sigma^2, \mathcal{V})$ with information cost $C$ and squared loss $R$, then either $C \geq \Omega\left(\frac{\sigma^2}{\delta^2 n \log(m)}\right)$ or $R \geq \delta^2/10$.*

## 4.2 Proof sketch of theorem 3.3

The protocol is described in protocol 3 in the supplementary. We only describe the $d = 1$ case, while for general case we only need to run $d$ protocols individually for each dimension.

The central idea is that we maintain an upper bound $U$ and lower bound $L$ for the target mean, and iteratively ask the machines to send their sample means to shrink the interval $[L, U]$. Initially we only know that $\theta \in [-1, 1]$. Therefore we set the upper bound $U$ and lower bound $L$ for $\theta$ to be $-1$ and 1. In the first iteration the machines try to determine whether $\theta < 0$ or $\geq 0$. This is done by letting several machines (precisely, $O(\log m)/\sigma^2$ machines) send whether their sample means are $< 0$ or $\geq 0$. If the majority of the samples are $< 0$, $\theta$ is likely to be $< 0$. However when $\theta$ is very close to 0, one needs a lot of samples to determine this, but here we only ask $O(\log m)/\sigma^2$ machines to send their sample means. Therefore we should be more conservative and we only update the interval in which $\theta$ might lie to $[-1, 1/2]$ if the majority of samples are $< 0$.

We repeat this until the interval $(L, U)$ become smaller than our target squared loss. Each round, we ask a number of new machines sending 1 bits of information about whether their sample mean is large than $(U + L)/2$. The number of machines participated is carefully set so that the failure probability $p$ is small. An interesting feature of the protocol is to choose the target error probability $p$ differently at each iteration so that we have a better balance between the failure probability and communication cost. The complete the description of the protocol and proof are given in the supplementary.

## 4.3 Proof sketch of theorem 3.4

We use a different prior on the mean $\mathcal{N}(0, \delta^2)$ instead of uniform over $\{-\delta, \delta\}$ used by [1]. Gaussian prior allows us to use a strong data processing inequality for jointly gaussian random variables by [14]. Since we don't have to truncate the gaussian, we don't lose the factor of $\log(m)$ lost by [1].

**Theorem 4.2.** *([14], Theorem 7) Suppose $X$ and $V$ are jointly gaussian random variables with correlation $\rho$. Let $Y \leftrightarrow X \leftrightarrow V$ be a markov chain with $I(Y; X) \leq R$. Then $I(Y; V) \leq \rho^2 R$.*

Now suppose that each machine gets $n$ samples $X^1, \ldots, X^n \sim \mathcal{N}(V, \sigma^2)$, where $V$ is the prior $\mathcal{N}(0, \delta^2)$ on the mean. By an application of theorem 4.2, we prove that if $Y$ is a $B$-bit message depending on $X^1, \ldots, X^n$, then $Y$ has only $\frac{n\delta^2}{\sigma^2} \cdot B$ bits of information about $V$. Using some standard information theory arguments, this converts into the statement that if $Y$ is the transcript of a simultaneous protocol with communication cost $\leq B$, then it has at most $\frac{n\delta^2}{\sigma^2} \cdot B$ bits of information about $V$. Then a lower bound on the communication cost $B$ of a simultaneous protocol estimating the mean $\theta \in [-1, 1]$ follows from proving that such a protocol must have $\Omega(1)$ bit of information about $V$. Complete proof is given in the supplementary.

# 5 Conclusion

We have lowerbounded the communication cost of estimating the mean of a $d$-dimensional spherical gaussian random variables in a distributed fashion. We provided a generic tool called direct-sum for relating the information cost of $d$-dimensional problem to one-dimensional problem, which might be of potential use for other statistical problem than gaussian mean estimation as well.

We also initiated the study of distributed estimation of gaussian mean with sparse structure. We provide a simple protocol that exploits the sparse structure and conjecture its tradeoff to be optimal:

**Conjecture 1.** *If some protocol estimates the mean for any distribution $P \in \mathcal{P}_s$ with mean-squared loss $R$ and communication cost $C$, then $C \cdot R \gtrsim \frac{sd\sigma^2}{mn}$, where we use $\gtrsim$ to hide log factors and potential corner cases.*

## Footnotes

[1]by minimax error we mean the minimum possible error that can be achieved when there is no limit on the communication

[2]technically, information cost, as discussed below

[3]where $I_d$ denote the $d \times d$ identity matrix

[4]Therefore here $\hat{\vec{\theta}}$ maps $\{0,1\}^*$ to $\Theta$

[5]Standard minimax theorem says that actually the $\sup_{\mathcal{V}^d} R_{\mathcal{V}^d}((\Pi, \hat{\vec{\theta}}), \vec{\theta}) = R((\Pi, \hat{\vec{\theta}}), \vec{\theta})$ under certain compactness condition for the space of $\vec{\theta}$.

[6]Note that here we have introduced a distribution for the choice of $\vec{\theta}$, and therefore $\vec{\theta}$ is a random variable.

[7] When $\sigma$ is very large, when $\theta$ is known to be in $[-1, 1]$, $\hat{\vec{\theta}} = 0$ is a better estimator, that is essentially why the lower bounds not only have the first term we desired but also the other two.

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
