[Supplementary Material]

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

# A  Communication Lower Bound via Direct-Sum Theorem: Proof of Theorem 3.1

We restate the main theorem here for convenience

**Theorem 3.1.** *[Direct-Sum] If* $(\Pi, \hat{\vec{\theta}})$ *solves the task* $T(d, m, n, \sigma^2, \mathcal{V}^d)$ *with information cost* $C$ *and squared loss* $R$*, then there exists* $(\Pi', \hat{\theta})$ *that solves the task* $T(1, m, n, \sigma^2, \mathcal{V})$ *with information cost at most* $4C/d$ *and squared loss at most* $4R/d$*. Furthermore, if the protocol* $\Pi$ *is simultaneous, then the protocol* $\Pi'$ *is also simultaneous.*

We consider the protocol $\Pi_i$ defined in Protocol 2. Lets denote the private and public randomness of the protocol $\Pi_i$ as $R_{\text{priv}}$ and $R_{\text{pub}}$ respectively. Note that in this section, $\theta$ is always a random variable from distribution $\mathcal{V}$ and $\vec{\theta}$ from $\mathcal{V}^d$. We skip the subscripts $\mathcal{V}$ and $\mathcal{V}^d$ when it is clear from the context.

Recall that we relate the information cost and mean-squared loss of $\Pi_i$'s and $\Pi$ by Lemma 1 and 2, which are restated and proved below.

**Lemma 1.** *Protocols* $\Pi_i$'s *satisfy* $\sum_{i=1}^{d} R_{\mathcal{V}} \left( (\Pi_i, \hat{\theta}_i), \theta \right) = R_{\mathcal{V}^d} \left( (\Pi, \hat{\vec{\theta}}), \vec{\theta} \right)$

**Lemma 2.** *Protocols* $\Pi_i$'s *satisfy* $\sum_{i=1}^{d} IC_{\mathcal{V}}(\Pi_i) \le IC_{\mathcal{V}^d}(\Pi)$

*Proof of Lemma 1.* The general idea is quite simple. By our design, the loss of each $\Pi_i$ is the loss of $\Pi$ restricted to the $i^{th}$ coordinate. The proof is an almost straightforward calculation that formalizes this intuition.

First note that by definition of the square loss and $\hat{\theta}_i$, we have

$$R_{\mathcal{V}} \left( (\Pi_i, \hat{\theta}_i), \theta \right) = \mathbb{E}[(\hat{\theta}_i(Y_i) - \theta)^2] = \mathbb{E}[(\hat{\vec{\theta}}(Y_i)_i - \theta)^2]$$

where the expectation over all the randomness of the mean, the data, and the protocols. Observe that under protocol $\Pi_i$, the distribution $(\breve{\theta}_{-i}, \theta)$ is $\mathcal{V}^d$ and therefore, the data $\breve{X}$ that machines prepared has the same distribution as $\vec{X}$. It follows that the joint distribution of $X, Y_i, (\theta, \breve{\theta}_{-i})$ is the same as the distribution of $\vec{X}_i, Y, \vec{\theta}$. Therefore,

$$\mathbb{E}[(\hat{\vec{\theta}}(Y_i)_i - \theta)^2] = \mathbb{E}[(\hat{\vec{\theta}}(Y)_i - \vec{\theta}_i)^2]] \tag{3}$$

Then it follows the linearity of expectation that

$$\sum_{i=1}^{d} R \left( (\Pi_i, \hat{\theta}_i), \theta \right) = \sum_{i=1}^{d} \mathbb{E}[(\hat{\vec{\theta}}(Y_i)_i - \theta)^2] = \sum_{i=1}^{d} \mathbb{E}[(\hat{\vec{\theta}}(Y)_i - \vec{\theta}_i)^2]]$$

$$= \mathbb{E}\left[ \sum_{i=1}^{d} (\hat{\vec{\theta}}(Y)_i - \vec{\theta}_i)^2 \right]$$

$$= \mathbb{E}[\|\hat{\vec{\theta}}(Y) - \vec{\theta}\|^2] = R_{\mathcal{V}^d} \left( (\Pi, \hat{\vec{\theta}}), \vec{\theta} \right)$$

where in the first line we used the definition and equation (3), the second line the linearity of expectation, the final line the definition again. $\square$

*Proof of Lemma 2.* Recall under $(\Pi_i, \hat{\theta}_i)$, machines prepare $\breve{X}$, which has the same distribution as $\vec{X}$ in the problem $T(d, m, n, \sigma^2, \mathcal{V}^d)$. Also the joint distribution of $\vec{X}_i, Y, \vec{\theta}$ is the same as the distribution of $X, Y_i, (\theta, \breve{\theta}_{-i})$. Therefore, we have that

$$I(\vec{X}_i; Y \mid \vec{\theta}) = I(X; Y_i \mid \theta, \breve{\theta}_{-i}) \tag{4}$$

By definition, $\text{IC}(\Pi_i) = I(X; Y_i \mid \theta, R_{\text{pub}})$, where $R_{\text{pub}}$ is $\breve{\theta}_{-i}$ because each machine publicly draws $\breve{\theta}_{-i}$ from $\mathcal{V}^{d-1}$. Therefore, $\text{IC}(\Pi_i) = I(X; Y_i \mid \theta, \breve{\theta}_{-i})$, and taking the sum over all $i$, and use equation (4)

$$\sum_{i=1}^{d} \text{IC}(\Pi_i) = \sum_{i=1}^{d} I(X; Y_i \mid \theta, \breve{\theta}_{-i})$$
$$= \sum_{i=1}^{d} I(\vec{X}_i; Y \mid \vec{\theta})$$

Note that the distribution of $\vec{X}$ conditioned on $\vec{\theta}$ is a spherical gaussian $\mathcal{N}(\vec{\theta}, \sigma^2 I_d)$, and recall that $\vec{X}_i$ is a shorthand for the collection of $i$th coordinates of all the samples: $\vec{X}_i = \{\vec{X}_i^{(j,k)} : j \in [m], k \in [n]\}$. Therefore, $\vec{X}_1, \ldots, \vec{X}_d$ are independent conditioned on $\vec{\theta}$. Hence,

$$\sum_{i=1}^{d} I(\vec{X}_i; Y \mid \vec{\theta}) \le I(\vec{X}; Y \mid \vec{\theta}) = \text{IC}(\Pi)$$

where the inequality follows Proposition E.1, a basic property of conditional mutual information.

$\square$

**Remark 3.** The role of private randomness can be crucially seen here. It is very important for the machines to privately get samples in coordinates other than $i$ for the information cost to go down by a factor of $d$.

*Proof of Theorem 3.1.* By Lemma 1 and Lemma 2 and a Markov argument, there exists an $i \in \{1, \ldots, d\}$ such that

$$R\left((\Pi_i, \hat{\theta}_i), \theta\right) \le \frac{4}{d} \cdot R\left((\Pi, \vec{\theta}), \vec{\theta}\right)$$

and

$$\text{IC}(\Pi_i) \le \frac{4}{d} \cdot \text{IC}(\Pi)$$

Then the pair $(\Pi', \hat{\theta}) = (\Pi_i, \hat{\theta}_i)$ solves the task $T(1, m, n, \sigma^2, \mathcal{V})$ with information cost at most $4C/d$ and squared loss $4R/d$. $\square$

We are going to apply the theorem above to the one-dimensional lower bound by [1]. Theorem A.1 below, though not explicitly stated, is implicit in the proof of Theorem 1 of [1]. Furthermore, their techniques are general enough to prove lower bounds on the information cost for protocols with private randomness, though they didn't mention this explicitly. Also in [1], the definition of information cost is a bit different. They do not condition on the prior of $\theta$, but since in the one dimensional case, this prior is just over $\{\pm\delta\}$, conditioning on it can reduce the mutual information by at most 1 bit.

$$I(X; Y \mid \theta, R_{\text{pub}}) \ge I(X; Y \mid R_{\text{pub}}) - H(\theta) \ge I(X; Y \mid R_{\text{pub}}) - 1$$

**Theorem A.1.** *[1] Let $\mathcal{V}$ be the uniform distribution over $\{\pm\delta\}$, where $\delta^2 \le \min\left(1, \frac{\sigma^2 \log(m)}{n}\right)$. If $(\Pi, \hat{\theta})$ solves the task $T(1, m, n, \sigma^2, \mathcal{V})$ with information cost $C$ and squared loss $R$, then either $C \ge \Omega\left(\frac{\sigma^2}{\delta^2 n \log(m)}\right)$ or $R \ge \delta^2/10$.*

The corollary below directly follows from Theorem A.1 and Theorem 3.1.

**Corollary A.1.** *Let $\mathcal{V}$ be the uniform distribution over $\{\pm\delta\}$, where $\delta^2 \le \min\left(1, \frac{\sigma^2 \log m}{n}\right)$. If $(\Pi, \hat{\theta})$ solves the task $T(1, m, n, \sigma^2, \mathcal{V}^d)$ with information cost $C$ and squared loss $R$, then either $C \ge \Omega\left(\frac{d\sigma^2}{\delta^2 n \log m}\right)$ or $R \ge d\delta^2/40$.*

Then noting that the communication cost is always larger than information cost, we can simply convert Corollary A.1 into lower bound for communication cost, Corollary 3.1, restated below for convenience.

**Corollary 3.1.** *Suppose* $(\Pi, \hat{\vec{\theta}})$ *estimates the mean of* $\mathcal{N}(\vec{\theta}, \sigma^2 I_d)$, *for all* $\vec{\theta} \in [-1, 1]^d$, *with mean-squared loss* $R$, *and communication cost* $B$. *Then*

$$R \geq \Omega\left(\min\left\{\frac{d^2\sigma^2}{nB\log m}, \frac{d\sigma^2}{n\log m}, d\right\}\right)$$

*As a corollary, when* $\sigma^2 \leq mn$, *to achieve the mean-squared loss* $R = \frac{d\sigma^2}{mn}$, *the communication cost* $B$ *is at least* $\Omega\left(\frac{dm}{\log m}\right)$.

*Proof.* Denote information cost of $(\Pi, \hat{\vec{\theta}})$ by $C$, and we have the trivial inequality $C \leq B$. The rest of proof concerns only about how to choose the right prior $\delta$ and to convert the bounds on $C$ and $R$ in Corollary A.1 into a single nice formula here. In the most typical case, if we choose $\delta^2 = \Omega(\frac{d\sigma^2}{nB\log n})$, it follows Corollary A.1 that

$$R \geq d\delta^2/40 \geq \Omega\left(\frac{d^2\sigma^2}{nB\log m}\right)$$

which captures the first term on the right hand side that we desired.

However, there are several corner cases that require additional treatment. Formally, we divide into two cases depending on whether $B \geq \frac{1}{c} \cdot \max\left(\frac{d\sigma^2}{n\log m}, \frac{d}{\log^2 m}\right)$ or not, where $c > 1$ is a constant to be specified later.

If $B \geq \frac{1}{c} \cdot \max\left(\frac{d\sigma^2}{n\log m}, \frac{d}{\log^2 m}\right)$, choose $\delta^2 = \frac{1}{c} \cdot \frac{d\sigma^2}{nB\log m}$. We can check $\delta^2 \leq \min\left(1, \frac{\sigma^2\log m}{n}\right)$, therefore we are ready to apply Corollary A.1. By the definition of $\delta$, we can check $C \leq B = \frac{1}{c} \cdot \frac{d\sigma^2}{\delta^2 n\log m}$. Choose $c$ large enough such that this violates the lower bound $C = \Omega(\frac{d\sigma^2}{\delta^2 n\log m})$ in Corollary A.1. Therefore, the other possible outcome of Corollary A.1 must be true, that is,

$$R \geq d\delta^2/40 \geq \Omega\left(\frac{d^2\sigma^2}{nB\log m}\right)$$

.

On the other hand, if $B \leq \frac{1}{c} \cdot \max\left(\frac{d\sigma^2}{n\log m}, \frac{d}{\log^2 m}\right)$, choose $\delta^2 = \frac{d\sigma^2}{n\max\left(\frac{d\sigma^2}{n\log m}, \frac{d}{\log^2 m}\right)\log m}$. Again $\delta^2 \leq \min\left(1, \frac{\sigma^2\log m}{n}\right)$ and by the definition of $\delta$,

$$C \leq B \leq \frac{1}{c} \cdot \max\left(\frac{d\sigma^2}{n\log m}, \frac{d}{\log^2 m}\right) = \frac{1}{c} \cdot \frac{d\sigma^2}{\delta^2 n\log m}$$

Hence $R \geq d\delta^2/40 \geq \Omega\left(\min\left\{\frac{d\sigma^2}{n\log m}, d\right\}\right)$.

Combining the two cases, we get

$$R \geq \Omega\left(\min\left\{\frac{d^2\sigma^2}{nB\log m}, \frac{d\sigma^2}{n\log m}, d\right\}\right)$$

$\square$

## B   Proof of Theorem 3.2

Let $S = \text{supp}(\vec{\theta})$. By sparsity of $\vec{\theta}$, we have $|S| \leq s$. For each $i \notin S$,

$$\mathbb{E}[(\hat{\theta}_i - \vec{\theta}_i)^2] = \mathbb{E}[\hat{\theta}_i^2] = \Pr[|\bar{X}_i|^2 > \alpha\sigma^2/(mn)]\,\mathbb{E}[\bar{X}_i^2 \mid |\bar{X}_i|^2 > \alpha\sigma^2/(mn)] < o(1/d^2) \cdot \frac{\alpha\sigma^2}{mn}$$

The last inequality follows the fact that the distribution of $\bar{X}_i$ is $\mathcal{N}(0, \frac{\alpha\sigma^2}{mnL\log d})$.

For any $i \in S$, we know that $\hat{\vec{\theta}}_i \in \{\bar{X}_i, 0\}$, therefore,

$$\mathbb{E}[(\hat{\vec{\theta}}_i - \vec{\theta}_i)^2] \leq \mathbb{E}[(\bar{X}_i - \vec{\theta}_i)^2 \mid \hat{\vec{\theta}}_i = \bar{X}_i]\Pr[\hat{\vec{\theta}}_i = \bar{X}_i] + \vec{\theta}_i^2 \Pr[\hat{\vec{\theta}}_i = 0]$$

The first term in RHS can be bounded by

$$\mathbb{E}[(\bar{X}_i - \vec{\theta}_i)^2 \mid \hat{\vec{\theta}}_i = \bar{X}_i]\Pr[\hat{\vec{\theta}}_i = \bar{X}_i] \leq \mathbb{E}[(\bar{X}_i - \vec{\theta}_i)^2] \leq \frac{\alpha\sigma^2}{mn}$$

For the second term, assuming wlog $\vec{\theta}_i > 0$, it is equal to $\vec{\theta}_i^2 \, \Phi(\left(\vec{\theta}_i - \sqrt{\frac{\alpha\sigma^2}{mn}}\right) \cdot \sqrt{\frac{Lmn\log d}{\alpha\sigma^2}})$,

which is upper bounded by $O(\frac{\alpha\sigma^2}{mn})$ when $L$ is sufficiently large constant.

Therefore, when $i \in S$, we have that $\mathbb{E}[(\hat{\vec{\theta}}_i - \vec{\theta}_i)^2] \leq O(\frac{\alpha\sigma^2}{mn})$. Putting all dimensions together,

$$\mathbb{E}[||\hat{\vec{\theta}} - \vec{\theta}||^2] = \sum_{i \in S}\mathbb{E}[(\hat{\vec{\theta}}_i - \vec{\theta}_i)^2] + \sum_{i \notin S}\mathbb{E}[(\hat{\vec{\theta}}_i - \vec{\theta}_i)^2] \leq O\left(\frac{\alpha s\sigma^2}{mn}\right)$$

Finally, the communication cost is clearly $O((dm\log m\log d)/\alpha)$ since totally $O((m\log d)/\alpha)$ $d$-dimensional vectors have been communicated.

## C Improved upper bound: proof of theorem 3.3

---

**Inputs** : Machine $j$ gets samples $X^{(j,1)}, \ldots, X^{(j,n)}$ distributed according to $\mathcal{N}(\theta, \sigma^2)$, where $\theta \in [-1, 1]$.

Each machine calculates its sample mean $\bar{X}^{(j)} = (X^{(j,1)} + \cdots + X^{(j,n)})/n$
The fusion center maintains global variables $L, U, \ell, p$ and broadcasts them if they are updated.
Initially, $U \leftarrow 1, L \leftarrow -1, \ell \leftarrow 0, p = 0.1m^{-3/2}$

**While** $U - L \geq 1/\sqrt{m}$

- $a \leftarrow (U + L)/2$
- Each machine $j \in \{\ell + 1, \ell + 1, \ldots, \ell + \frac{50\log(2/p)}{\sigma^2(U-L)^2}\}$ sends whether $m^j = 1$ if $\bar{X}^{(j)} \geq a$ otherwise 0.
- If the majority of $m^j$ for $j \in \{\ell + 1, \ell + 1, \ldots, \ell + \frac{50\log(2/p)}{\sigma^2(U-L)^2}\}$ is 1, then $L \leftarrow (L + a)/2$. Otherwise $U \leftarrow (U + a)/2$.
- $\ell \leftarrow \ell + \frac{50\log(1/p)}{\sigma^2(U-L)^2}, p = p \cdot \left(\frac{4}{3}\right)^3$.

**end**

**Output** $L$

---

**Protocol 3:** Improved Interactive Protocol for One-dimensional Gaussian Mean Estimation

For simplicity, and without loss of generality, we only prove the case when $n = 1$ and $\sigma = 1$. In this case, each machine gets one sample from $\mathcal{N}(\theta, 1)$. Our goal is to prove that Protocol 3 has communication cost $O(m)$ and mean-squared loss $O(1/m)$

Before going into the proof, we provide some justification for making the error probability of each round exponentially decreasing. Intuitively, when the interval $[L, U]$ is small, we may allow slightly larger failure probability since even we fail, the squared loss caused won't be large given $[L, U]$ is small. It turns out the right tradeoff is to increase the error probabilities exponentially as the

approximation of $\theta$ gets better for two reasons: 1) the squared loss is affected more if the protocol fails early when the estimate is still coarse so we want the failure probability in the early iteration to be very small 2) the number of samples needed for the coarse approximation is small so it is cheaper to decrease the failure probability of the early iterations than that of the late iterations.

Let $\Phi(x)$ be the c.d.f for normal distribution $\mathcal{N}(0,1)$. We will need the following simple lemma $\Phi(x)$ which is essentially the fact that the p.d.f of normal distribution is close to a constant around 0. We delay the proof of the lemma to the end of the section.

**Lemma 3.** *For $0 \leq t \leq 1$, we have $\Phi(t) \geq 1/2 + t/4$.*

Note that initially $U - L = 2$ and in each iteration, $U - L$ decreases by a factor of $3/4$, therefore the number of iterations is at most $T = \log_{4/3}(2\sqrt{m})$. Let $U_0 = 1$, $L_0 = -1$ and $U_s, L_s$ be the value of $U$ and $L$ after $s$ iterations, and let $t_s = U_s - L_s$. Also denote the value of $p$ after $s$ iterations as $p_s$. Therefore, by the definition of the protocol, $t_s = 2 \cdot (3/4)^s$ and $p_s = (4/3)^{3s} \cdot 0.1 m^{-3/2}$.

We thought $p_s$ a the failure probability we would like to tolerate for iteration $s$. We make this formal by defining $E_s$ be the indicator variable for the event that $\theta \in [L_s, U_s]$, that is, the event that the protocol outputs a valid interval that contains $\theta$ after $s$ iteration. We claim that

**Claim 1.** $\Pr[E_{s+1} = 0 | E_s = 1] \leq p_s$

*Proof Of claim 1.* Assuming $E_s$ happens, we know that $\theta \in [L_s, U_s]$. If $E_{s+1}$ doesn't happen, then there must be two cases: a) $\theta \in [L_s, (3L_s + U_s)/4]$, and the majority of the $m^j$'s at that iteration is 1. b) $\theta \in [(L_s + 3U_s)/4, U_s]$, and the majority of the $m^j$'s at that iteration is 0. These two cases are symmetric and we only analyze the first one. Under case a), the probability that a single gussian sample from $\mathcal{N}(\theta, 1)$ is less than $a = (U_s + L_s)/2$ is $1 - \Phi(t_s/4) \leq 1/2 - t_s/20$. Therefore by chernoff bound, probability that majority of $t$ independent samples from $\mathcal{N}(\theta, 1)$ are greater than $(L_s + U_s)/2$ is $\leq e^{-t \cdot t_s^2/50}$. In the protocol, we have $t = 50 t_s^2 \cdot \log(2/p_s)$ and hence $e^{-t \cdot t_s^2/50} \leq p_s/2$. $\qquad\square$

Then let's calculate the mean-squared loss and the communication cost. For squared loss, let $s$ be the smallest $s$ such that $E_{s+1} = 0$. In this case, the squared loss is at most $t_s^2$ since we know $\theta \in [L_s, U_s]$ and the final output will also be in this interval. Note that $\Pr[E_s = 1, E_{s+1} = 0] \leq \Pr[E_{s+1} = 0 \mid E_s = 1] \leq p_s$ by Claim 1, therefore the expected square loss is at most

$$\text{total squared loss} \leq \sum_{s=0}^{T} q_s t_s^2 = \sum_{s=0}^{T} \left(\frac{4}{3}\right)^{3s} \cdot 1/10 m^{3/2} \cdot 4 \cdot \left(\frac{3}{4}\right)^{2s}$$

$$= \frac{4}{10 m^{3/2}} \cdot \sum_{s=0}^{T} \left(\frac{4}{3}\right)^{s}$$

$$= O(1/m)$$

The total communication is simply

$$50 \cdot \sum_{s=0}^{T} t_s^2 \cdot \log(1/q_s) = O\left(\sum_{s=0}^{T} \left(\frac{4}{3}\right)^{2s} \cdot \log\left(\left(\frac{3}{4}\right)^{3s} \cdot 10 m^{3/2}\right)\right)$$

$$= O\left(\sum_{s=0}^{T} \left(\frac{4}{3}\right)^{2s} \cdot \log\left(10/8 \cdot \left(\frac{4}{3}\right)^{T-s}\right)\right)$$

$$= O\left(\sum_{s=0}^{T} \left(\frac{4}{3}\right)^{T-s} \cdot \log\left(10/8 \cdot \left(\frac{4}{3}\right)^{s}\right)\right)$$

$$= O(m)$$

The third equality is just a change of variable. The fourth equality follows from the fact that $\sum_{s=0}^{\infty} \left(\frac{3}{4}\right)^{s} \cdot s = O(1)$. Note that we have used $O(m)$ samples whereas we have only $m$ machines, but we can just increase $m$ by a constant factor, thereby incurring another constant factor in the expected square loss.

*Proof of Lemma 3.*

$$\frac{1}{\sqrt{2\pi}} \int_{-\infty}^{t} e^{-x^2/2}\,\mathrm{d}x = \frac{1}{\sqrt{2\pi}} \int_{-\infty}^{0} e^{-x^2/2}\,\mathrm{d}x + \frac{1}{\sqrt{2\pi}} \int_{0}^{t} e^{-x^2/2}\,\mathrm{d}x$$

$$= 1/2 + \frac{1}{\sqrt{2\pi}} \int_{0}^{t} e^{-x^2/2}\,\mathrm{d}x$$

$$\geq 1/2 + \frac{1}{\sqrt{2\pi}} \int_{0}^{t} (1 - x^2/2)\,\mathrm{d}x$$

$$= 1/2 + \frac{1}{\sqrt{2\pi}} \left(t - t^3/6\right)$$

$$\geq 1/2 + t/4$$

$\square$

# D  Improved lower bound: Proof of theorem 3.4

We will need the following theorem from [14].

**Theorem D.1.** *([14], Theorem 7) Suppose $X$ and $V$ are jointly gaussian random variables with correlation $\rho$. Let $Y \leftrightarrow X \leftrightarrow V$ be a markov chain with $I(Y;X) \leq R$. Then $I(Y;V) \leq \rho^2 R$.*

We prove a slight generalization of the above theorem which we'll need for our lower bound.

**Lemma 4.** *Suppose $V \sim \mathcal{N}(0, \delta^2)$. Let $Z_1, \ldots, Z_n$ be iid gaussians with mean $0$ and variance $\sigma^2$, and $X^i = V + Z_i$. If $Y \leftrightarrow X^1, \ldots, X^n \leftrightarrow V$ is a markov chain s.t. $I(Y; X^1, \ldots, X^n) \leq R$, then $I(Y;V) \leq \frac{n\delta^2}{\sigma^2 + n\delta^2} R$.*

*Proof.* Consider the density of $v$ conditioned on $x^1, \ldots, x^n$. Let $\bar{x} = \sum_{i=1}^{n} x^i$.

$$p(v|x^1, \ldots, x^n) = \frac{e^{-v^2/2\delta^2} \cdot e^{-\sum_{i=1}^{n}(x^i - v)^2/2\sigma^2}}{\int_{-\infty}^{\infty} e^{-v^2/2\delta^2} \cdot e^{-\sum_{i=1}^{n}(x^i - v)^2/2\sigma^2}\,\mathrm{d}v}$$

$$= \frac{e^{-v^2/2\delta^2} \cdot e^{\bar{x}v/\sigma^2 - nv^2/2\sigma^2}}{\int_{-\infty}^{\infty} e^{-v^2/2\delta^2} \cdot e^{\bar{x}v/\sigma^2 - nv^2/2\sigma^2}\,\mathrm{d}v}$$

$$= \frac{e^{-v^2/2\delta^2} \cdot e^{\bar{x}v/\sigma^2 - nv^2/2\sigma^2}}{e^{\frac{\bar{x}^2 \delta^2}{2\sigma^2(\sigma^2 + n\delta^2)}} \cdot \int_{-\infty}^{\infty} e^{-(v - \frac{\bar{x}\delta^2}{\sigma^2 + n\delta^2})/2 \frac{\delta^2\sigma^2}{\sigma^2 + n\delta^2}}\,\mathrm{d}v}$$

$$= \frac{e^{-(v - \frac{\bar{x}\delta^2}{\sigma^2 + n\delta^2})/2 \frac{\delta^2\sigma^2}{\sigma^2 + n\delta^2}}}{\int_{-\infty}^{\infty} e^{-(v - \frac{\bar{x}\delta^2}{\sigma^2 + n\delta^2})/2 \frac{\delta^2\sigma^2}{\sigma^2 + n\delta^2}}\,\mathrm{d}v}$$

Thus the distribution of $v|x^1, \ldots, x^n$ is $\mathcal{N}(\frac{\bar{x}\delta^2}{\sigma^2 + n\delta^2}, \frac{\delta^2\sigma^2}{\sigma^2 + n\delta^2})$, and hence also the distribution of $v|\bar{x}$ is $\mathcal{N}(\frac{\bar{x}\delta^2}{\sigma^2 + n\delta^2}, \frac{\delta^2\sigma^2}{\sigma^2 + n\delta^2})$. Moreover, the distribution of $\bar{x}$ is $\mathcal{N}(0, n(\sigma^2 + n\delta^2))$ and hence of $\frac{\bar{x}\delta^2}{\sigma^2 + n\delta^2}$ is $\mathcal{N}(0, \frac{n\delta^4}{\sigma^2 + n\delta^2})$. Hence $V$ and $\frac{(\sum_{i=1}^{n} X^i)\delta^2}{\sigma^2 + n\delta^2}$ are jointly gaussian random variables with correlation $\rho = \frac{\sqrt{n\delta^2}}{\sqrt{\sigma^2 + n\delta^2}}$. Also $Y \leftrightarrow X^1, \ldots, X^n \leftrightarrow \frac{(\sum_{i=1}^{n} X^i)\delta^2}{\sigma^2 + n\delta^2} \leftrightarrow V$ is a markov chain. Data processing implies that $I(Y; \frac{(\sum_{i=1}^{n} X^i)\delta^2}{\sigma^2 + n\delta^2}) \leq I(Y; X^1, \ldots, X^n) \leq R$. Hence applying theorem D.1, we get that $I(Y;V) \leq \frac{n\delta^2}{\sigma^2 + n\delta^2} R$. $\square$

An easy corollary is the following:

**Corollary D.1.** *Suppose $V \sim \mathcal{N}(0, \delta^2)$. Let $Z_1, \ldots, Z_n$ be iid gaussians with mean $0$ and variance $\sigma^2$, and $X^i = V + Z_i$. If $Y \leftrightarrow X^1, \ldots, X^n \leftrightarrow V$ is a markov chain, then $I(Y;V) \leq \frac{n\delta^2}{\sigma^2} \cdot I(Y; X^1, \ldots, X^n | V)$.*

*Proof.* Since $Y \leftrightarrow X^1, \ldots, X^n \leftrightarrow V$ is a markov chain, $I(Y; X^1, \ldots, X^n | V) = I(Y; X^1, \ldots, X^n) - I(Y; V)$. Since by lemma 4, $I(Y; X^1, \ldots, X^n) \geq \frac{\sigma^2 + n\delta^2}{n\delta^2} \cdot I(Y; V)$, we get $I(Y; X^1, \ldots, X^n | V) \geq \frac{\sigma^2}{n\delta^2} I(Y; V)$, or $I(Y; V) \leq \frac{n\delta^2}{\sigma^2} \cdot I(Y; X^1, \ldots, X^n | V)$. $\qquad\square$

This leads to the following lemma:

**Lemma 5.** *If* $\Pi$ *is a simultaneous protocol for* $m$ *machines, where machine* $i$ *gets* $n$ *samples* $X^{(i,1)}, \ldots, X^{(i,n)} \sim \mathcal{N}(V, \sigma^2)$, *where* $V \sim \mathcal{N}(0, \delta^2)$. *Then the information cost of the protocol* $\Pi$, $I$ *satisfies* $I(Y; V) \leq \frac{n\delta^2}{\sigma^2} \cdot I$, *where* $Y$ *is the transcript of the protocol* $\Pi$.

*Proof.* Since $\Pi$ is a simultaneous protocol, machine $i$ sends a message $Y^i$ based on $X^{(i,1)}, \ldots, X^{(i,n)}$. Suppose $X^i$ denote $X^{(i,1)}, \ldots, X^{(i,n)}$. Then by corollary D.1, we have that $I(Y^i; V) \leq \frac{n\delta^2}{\sigma^2} \cdot I(Y^i; X^i | V)$. The information cost of the protocol $\Pi$ is $I(Y^1, \ldots, Y^n; X^1, \ldots, X^n | V)$. Note that $(Y^1, X^1), \ldots, (Y^n, X^n)$ are independent conditioned on $V$. This gives us:

$$I = I(Y^1, \ldots, Y^n; X^1, \ldots, X^n | V)$$
$$= \sum_{i=1}^{n} I(Y^i; X^i | V)$$
$$\geq \frac{\sigma^2}{n\delta^2} \cdot \sum_{i=1}^{n} I(Y^i; V)$$

To complete the proof of the lemma, we need to prove that $\sum_{i=1}^{n} I(Y^i; V) \geq I(Y; V)$, which follows from proposition E.1.

$\qquad\square$

Now we have the tools to prove theorem D.2 about improved lower bound for gaussian mean estimation for simultaneous protocols.

**Theorem D.2.** *Suppose* $(\Pi, \hat{\theta})$ *estimates the mean of* $\mathcal{N}(\theta, \sigma^2)$, *for all* $\theta \in [-1, 1]$, *with mean-squared loss* $R$, *and communication cost* $B$, *where* $\Pi$ *is a simultaneous protocol. Then*

$$R \geq \Omega\left(\min\left\{\frac{\sigma^2}{nB}, 1\right\}\right)$$

*As a corollary, to achieve the optimal mean-squared loss* $R = \frac{\sigma^2}{mn}$, *the communication cost* $B$ *is at least* $\Omega(m)$.

*Proof.* We can assume $R \leq 1/100$, otherwise we are done. Consider a simulation of the protocol $\Pi$ where the mean $\theta$ is generated according to the distribution $\mathcal{N}(0, \delta^2)$, where $\delta$ will be chosen appropriately. We'll denote by $V$, the random variable for the mean. If $Y$ denotes the transcript of the protocol, then by lemma 5, we have $I(Y; V) \leq \frac{n\delta^2}{\sigma^2} \cdot B$ (since information cost is upper bounded by communication cost). Let $S$ be the sign of $V$. Also let $\delta^2 = 10R$. Then since square loss of the estimator $\hat{\theta}(Y)$ is $R$, using $Y$, one can predict $S$ w.p. $1/2 + \Omega(1)$ (with the predictor sign($\hat{\theta}(Y)$)). Hence $I(Y; S) \geq \Omega(1)$ (e.g. by Fano's inequality), which implies $I(Y; V) \geq \Omega(1)$ (by data processing). Hence $\frac{n\delta^2}{\sigma^2} \cdot B \geq \Omega(1)$, which implies $R \geq \Omega\left(\frac{\sigma^2}{nB}\right)$. $\qquad\square$

The proof of theorem 3.4 is an easy application of the direct sum theorem (theorem 3.1), lemma 5, and arguments similar to the proof of theorem D.2, so we skip it.

# E   Information Theory Inequalities

**Proposition E.1.** *If random variables $\vec{X}_1, \ldots, \vec{X}_d$ are independent conditioned on the random variable $\vec{\theta}$, then for any random variable $Y$, we have,*

$$\sum_{i=1}^{d} I(\vec{X}_i; Y \mid \vec{\theta}) \leq I(\vec{X}_1 \ldots \vec{X}_d; Y \mid \vec{\theta})$$

*Proof.* We first use the chain rule for condition information and get

$$
\begin{aligned}
I(\vec{X}; Y \mid \vec{\theta}) &= \sum_{i=1}^{d} I(\vec{X}_i; Y \mid \vec{\theta}, \vec{X}_1, \ldots, \vec{X}_{i-1}) \\
&= \sum_{i=1}^{d} \left( H(\vec{X}_i \mid \vec{\theta}, \vec{X}_1, \ldots, \vec{X}_{i-1}) - H(\vec{X}_i \mid Y, \vec{\theta}, \vec{X}_1, \ldots, \vec{X}_{i-1}) \right)
\end{aligned}
$$

Then since $\vec{X}_1, \ldots, \vec{X}_d$ are independent conditioned on $\vec{\theta}$, we have $H(\vec{X}_i \mid \vec{\theta}, \vec{X}_1, \ldots, \vec{X}_{i-1}) = H(\vec{X}_i \mid \vec{\theta})$, and then

$$
\begin{aligned}
I(\vec{X}; Y \mid \vec{\theta}) &= \sum_{i=1}^{d} \left( H(\vec{X}_i \mid \vec{\theta}) - H(\vec{X}_i \mid Y, \vec{\theta}, \vec{X}_1, \ldots, \vec{X}_{i-1}) \right) \\
&\geq \sum_{i=1}^{d} \left( H(\vec{X}_i \mid \vec{\theta}) - H(\vec{X}_i \mid Y, \vec{\theta}) \right) \\
&= \sum_{i=1}^{d} I(\vec{X}_i; Y \mid \vec{\theta})
\end{aligned}
$$

where the inequality follows from the fact that conditioning decreases entropy. $\quad\square$