[Reviews · NeurIPS 2014]

Submitted by Assigned_Reviewer_13

This paper investigates the communication cost of distributed estimation for d-dimensional spherical Gaussian distribution with unknown mean and unitary covariance, where the joint distribution is assumed to be a product distribution of each coordinate. The authors generalize previous works on the one-dimensional case in [4] by proposing upper and lower bounds for d-dimensional data on two communication schemes, interactive and simultaneous communication settings, for achieving minimax squared loss. The results establish the tradeoffs between dimensionality and communication cost for distributed estimation. In addition, improved bounds are derived when the unknown mean is s-sparse.

Although the model is somewhat restrictive (for example, the joint distribution is a product distribution of each coordinate and the covariance matrix is known), this paper offers new insights on the communication cost of distributed estimation and protocol design. The results are technically solid and several rule-of-thumb protocols have been proposed with an aim of achieving the communication lower bounds.

A minor concern for this reviewer was the paper's weakness in presentation. First, the introduction is too lengthy and, second, there is no conclusion section. It is suggested that the authors refine the introduction part and provide a conclusion section to summarize the contributions. It was also somewhat irritating that the notations in the main paper and the supplementary file were not consistent, making it difficult to use the supplementary file for clarification of certain points in the paper. For example, Lemmas 1 and 2 in the paper are exactly the same as Lemmas 3 and 4 in the supplementary file.
Summary: This paper studies bounds on communication cost in distributed estimation and several protocols are proposed to achieve the communication lower bounds. The authors provide solid analysis on the subject and the results are significant.

Submitted by Assigned_Reviewer_46

This paper explores the connection between dimensionality and communication cost in distributed learning problems. In this problem, the samples from the unknown distribution are distributed among m different machines. The goal is to estimate the mean at the optimal minimax rate while communicating as few bits as possible. This paper shows that in this setting, the communication cost scales linearly in the number of dimensions. Applying this result to previous lower bounds for one dimension in the interactive setting [4] and to the improved bounds for the simultaneous setting, this paper proves new lower bounds of \Omega(md/ log(m)) and \Omega(md) for the bits of communication needed to achieve the minimax squared loss, in the interactive and simultaneous settings respectively. Besides, this paper demonstrates an interactive protocol achieving the minimax squared loss with O(md) bits of communication. Given the strong lower bounds in the general setting, this paper initiates the study of the distributed parameter estimation problems with structured parameters. Specifically, when the parameter is known to be s-sparse, it shows a protocol achieving the minimax squared loss with high probability and with communication cost proportional to s rather than the dimension d of the ambient space.

Quality: The authors need to assume that the readers have no knowledge on the problem in [4].

Clarity: In Section 2, definitions are missing for several notions:
0. What is the definition of s-sparse?
1. In the definition of R(\hat{\theta},\theta), why the expectaion is over \hat{\theta},X,\theta? How is the mean \vec{\theta}(Y) of Y obtained? Why is Y in {\cal X}^n? Is there any example for transcript?
2. I cannot follow below Private/Public Randomness. For example,
a. What is private and public randomness? What is a protocol? Give definitions to them.
b. Why can the public randomness be shared among the machines before the start of the protocol?
c. Why does the protocol work well on average over all public randomness?
d. Why can the machine use private randomness to hide information from other machines in a protocol?
e. The definition ``(\Pi,\vec{\theta}) solves T(d,m,n,\sigma^2,{\cal D}_\theta^d) with C and R.

Oiginality: I believe that new lower and upper bounds are derived for the interactive and simultaneous settings, and that the negative result is obtained. It is fair that the authors explain how difficult the derivation fom [4] is.

Significance: I do not understand why ''the communication cost scales linearly in the number of dimensions" is a news. In fact, ituitively, this seems to be obvious while they mathematically derived the property. The authors need to explain why the result is different from what they expected.
Summary: The authors should assume that the readers are not familiar with the material. Some introductory explanation could be used. I admit that the idea is novel.

Submitted by Assigned_Reviewer_47

This paper consider the problem of estimating the mean of a d-dimensional normal distribution with unit covariance, given nm samples available at m machines. The authors study the trade-off between communication complexity and dimensionality.
Several results are derived:
- Communication complexity cannot be reduced by processing jointly the various dimensions.
- It can be reduced for sparse means.
- Improved upper bound in the scalar case.
- Improved lower bound under simultaneous protocols.

I have a few minor suggestions:
1) At the beginning \theta is not a random variable, but it becomes a random variable after Definition 1. This is a standard device in minimax theory, but it is worth emphasizing the passage. Also, before definition 1, the authors introduce the conditional mutual information given theta. Strictly speaking makes sense only for theta a random variable (although taking a constant value with probability 1).

2) In Corollary 3.1. The minimax rate for the stated problem is not d\sigma^2/(nm). Thing of the case in which \sigma^2 = 1000000, n = m = d = 1. Then the estimator \hat{\theta} = 0 can do better than that. The statement must be qualified.

3) In Protocol 2, last two lines. I believe that Y_i in the argument of \hat{\theta}_i should not have subscript.
Summary: A very nice papers with several interesting ideas.
Author Feedback
Author rebuttal: First off, we would like to thank the reviewers for very insightful comments and suggestions. We will revise the write-up to improve the introduction, provide a conclusion section, and fix the typos and numbering problems you pointed out.

In the limited space below, we will try to clarify the questions of Reviewer 2. We are going to add more background and further clarify these confusions in Section 2 of the paper.

0. What is the definition of s-sparse?

the vector has s non-zero coordinates

1. In the definition of R(\hat{\theta},\theta), why the expectaion is over \hat{\theta},X,\theta?

This is a typo. The expectation in the definition of R(\hat{\theta}, \theta) is over X and the randomness of the estimator \hat{\theta}. Later on when \theta is also considered as a random variable (like in the standard minimax argument), then the expectation is over all the randomness.

How is the mean \vec{\theta}(Y) of Y obtained? Why is Y in {\cal X}^n? Is there any example for transcript?

The transcript is the concatenation of the messages being communicated, which could be arbitrarily long, and is not in {\cal X}^n. Transcript is determined by the input X. In the multi-machine setting, \vec{\theta} takes the transcript Y as input, instead of the data X, and outputs the estimate for the mean \theta. Therefore, in the multi-machine setting, \vec{\theta} is a mapping from {0,1}^* to \Theta.

2. I cannot follow below Private/Public Randomness. For example,
a. What is private and public randomness? What is a protocol? Give definitions to them.

A protocol is an agreed upon sequence of interactions where in each round, a specified party sends a message to some other party. In our case, the message is written on a public board so that all parties can see it. Each party/machine has access to its own sequence of random bits, which is its private randomness. Public randomness is a sequence of random bits that is shared among all parties before the protocol without being counted toward the total communication.

b. Why can the public randomness be shared among the machines before the start of the protocol?

As mentioned in the paper, public randomness is used purely for convenience in proofs: for any protocol using public randomness, we can find a fixed random string with which the protocol performs as well as using public randomness. (Alternatively, using pseudo-random generator with a short seed is a quick solution in practice). Thus, it is used only to simplify description and is not a way to cheat.

c. Why does the protocol work well on average over all public randomness?

The loss of a protocol with public randomness is defined as the average loss over all choices of the public randomness.

d. Why can the machine use private randomness to hide information from other machines in a protocol?

Private randomness means that each machine draws random coins by itself. The results of these random coin tosses are not known by other machines. Consider an example where Alice has x, Bob has y, where 1<=x,y<=300 and they want to determine if either |x-y| <= 10 or |x-y| >= 100 (it is promised that x and y satisfy either of these conditions). One way to solve this is for Alice to send x to Bob but it reveals x to Bob. Instead, Alice could draw a random noise r in [-10,10] and send x+r to Bob. With x+r, Bob can resolve the question without knowing for certain what x is (because he doesn't know what r is). Thus, Alice used her private randomness (r) to hide information from Bob. Hiding information is only used for proving lower bounds, where in the reduction we want the machine to communicate less information so that it contradicts the one-dimensional lower bound.

e. The definition ``(\Pi,\vec{\theta}) solves T(d,m,n,\sigma^2,{\cal D}_\theta^d) with C and R.

It's just a short way of saying the protocol has square loss R and information cost C for the certain parameters d,m,n,\sigma^2 and the underlying distribution {\cal D}_{\theta^d}

3. I do not understand why ''the communication cost scales linearly in the number of dimensions" is a news

We agree it is indeed what intuition suggests. However, it is still a significant technical challenge and actually in some very related situations, the kind of intuition that “d-dimensional statistical problem is d times harder than 1-dimensional problem” could be misleading: the famous James-Stein estimator is an example of savings when estimating multiple dimensions together instead of separately. (Unfortunately we cannot explain this example in the limited space here but it can be found in standard statistics literature).